Metagenomic investigation of bacterial laccases in a straw-amended soil

Yu Dali yudali@qlnu.edu.cn 1 2
Liu Ying 1
Cai Hongying 2
Huang Wanqiu 2
Wu Huijun 3
Yang Peilong yangpeilong@caas.cn 2
1 Qilu Normal University , Jinan , Shandong , China
2 Key Laboratory for Feed Biotechnology of the Ministry of Agriculture and Rural Affairs, National Engineering Research Center of Biological Feed, Institute of Feed Research, Chinese Academy of Agricultural Sciences , Beijing , China
3 Institute of Agricultural Resources and Regional Planning, Chinese Academy of Agricultural Sciences , Beijing , China
Grohmann Elisabeth
Electronic publication date: 2025 Apr 28
Publication date: 2025
Volume: 13
Electronic Location ID: e19327
Received 2024 Oct 8; Accepted 2025 Mar 25
Copyright: ©2025 Yu et al.
Copyright year: 2025
Copyright holder: Yu et al.
License: This is an open access article distributed under the terms of the Creative Commons Attribution License, which permits unrestricted use, distribution, reproduction and adaptation in any medium and for any purpose provided that it is properly attributed. For attribution, the original author(s), title, publication source (PeerJ) and either DOI or URL of the article must be cited.
License URL: https://creativecommons.org/licenses/by/4.0/

Keywords: Bacterial laccase, Metagenomics, Straw-amended soil, Profile hidden Markov models

Funding: National Key Research and Development Program of China No. 22YFD1200601 Project of Shandong Province Higher Educational Science and Technology Program No. J18AK175 This work was supported by the National Key Research and Development Program of China (No. 22YFD1200601) and a Project of Shandong Province Higher Educational Science and Technology Program (No. J18AK175). The funders had no role in study design, data collection and analysis, decision to publish, or preparation of the manuscript.

==============================
Background

Bacterial laccases play a crucial role in the degradation of lignin and the turnover of soil organic matter. Their advantageous properties make them highly suitable for a wide range of industrial applications. However, the limited identification of these potential enzymes has impeded their full utilization. The straw-amended soil provides materials for the development of bacterial laccases.

Methods

Metagenomic sequencing of a straw-amended soil was conducted to explore novel bacterial laccases. The putative bacterial laccases were then screened using profile hidden Markov models for further analysis. The most abundant gene, lacS1, was heterologously expressed in Escherichia coli and the recombinant laccase was purified for enzymatic characterization.

Results

A total of 322 putative bacterial laccases were identified in the straw-amended soil. Among them, 45 sequences had less than 30% identity to any entries in the Carbohydrate-Active Enzyme database and only 4.66% were more than 75% similar to proteins in the NCBI environmental database, exhibiting their novelty. These enzymes were found across various bacterial orders, demonstrating substantial diversity. Phylogenetic analysis revealed a number of the bacterial laccase sequences clustered with homologs characterized by favorable enzymatic properties. Five full-length representative bacterial laccase genes were obtained by modified thermal asymmetric interlaced PCR. The laccase activity of lacS1 was validated. It was a mesophilic enzyme with alkaline stability and halotolerance, indicating its promise for industrial applications.

Implications

These findings highlight novel bacterial laccase resources with potential for industrial applications and enzyme engineering.

Introduction

Plant biomass constitutes the primary source of soil organic matter, playing a crucial role in carbon and nutrient cycling as it decomposes (Prescott, 2005; Poll et al., 2008). This decomposition process is predominantly driven by soil microorganisms, which break down plant litter—composed of both labile and recalcitrant compounds (Paterson et al., 2008). Among these compounds, recalcitrant substrates such as aromatic polymers are particularly resistant to microbial degradation. Specific microbial groups thrive only when simpler substrates are made available by other groups that specialize in the degradation of complex compounds, like lignin (McGuire & Treseder, 2010) As the second most abundant component of plant residues, lignin serves as a major source of these recalcitrant natural polymers (Wong, 2009; Theuerl & Buscot, 2010). Microorganisms produce several intra- and extracellular enzymes to catalyze the breakdown of these plant polymers, including lignin peroxidases, manganese peroxidases, versatile peroxidases, and laccases (Pollegioni, Tonin & Rosini, 2015). Among these enzymes, laccases are noted for forming the most aromatic-aromatic interactions and appear particularly effective in lignin deconstruction (De Angelis et al., 2011; Chen et al., 2011).

Laccases (benzendiol: oxygen oxidoreductase, EC 1.10.3.2), members of the multicopper oxidase (MCO) superfamily, are capable of oxidizing a wide array of phenolic compounds and certain non-phenolic substrates, often through the mediation of small molecules (Zhai et al., 2024). During this oxidation process, molecular oxygen serves as the electron acceptor and is reduced to water as the sole by-product. Due to these unique characteristics, laccases hold significant promise as environmentally friendly biocatalysts for various industrial applications, including textile dye decolorization, effluent detoxification, and paper and pulp production (Sodhi, Bhatia & Batra, 2024). Initially discovered in the sap of the Japanese lacquer tree (Rhus vernicifera), laccases have since been found to be widely distributed among fungi (Yoshida, 1883; Thurston, 1994). To date, most identified and characterized laccases originate from fungal sources, and fungal laccases are extensively used in industrial processes due to their relatively high activity.

Recently, however, increasing evidence has shown that laccases are also prevalent in bacteria (Alexandre & Zhulin, 2000; Claus, 2003). Compared to their fungal counterparts, bacterial laccases exhibit several advantageous properties, such as high thermal stability, chloride tolerance, and broad pH stability, which are required in harsh industrial environments (Liu et al., 2017; Sharma & Leung, 2021; Liu et al., 2023). Therefore, they were more suitable for industrial applications. Despite their potential, only a limited number of bacterial laccases have been thoroughly characterized, underscoring the need for further investigation into their properties for novel biotechnological applications.

In the context of sustainable agriculture, the incorporation of straw as an amendment is widely regarded as an effective and economical agricultural practice. Long-term straw amendment promotes the sequestration of soil organic carbon, thereby contributing to soil quality maintenance (Liu et al., 2014). Straw is primarily decomposed by soil microorganisms, and its incorporation into the soil can influence microbial community composition. Our previous research demonstrated that the soil bacterial communities were altered after six years of continuous straw return treatment (Yu et al., 2018). Given that the straw-amended soil serves as a unique habitat for lignocellulosic-degrading microbial consortia, it is likely to be rich in bacterial laccase resources. In previous studies, bacterial laccase genes have primarily been identified using pure culture techniques and metagenomic library approaches (Zhang et al., 2022; Fang et al., 2012). However, these methods are often time-consuming and inherently biased, limiting their effectiveness for making a general survey of the capacity of bacterial laccase genes. High-throughput sequencing offers a novel and unbiased approach to detect genes in environmental samples and can also assess their abundance and diversity. In this study, we comprehensively investigated the straw-amended soil through metagenomic sequencing to uncover novel laccases with potential for biotechnological applications.

Materials & Methods

Soil sampling and DNA extraction

Soil samples were collected from an experimental field located at the Agricultural Station of the Chinese Academy of Agricultural Sciences (39°36′N, 116°36′E), situated at an elevation of 18 m above sea level in Langfang, Hebei, China. The experiment site, soil taxonomy and soil properties were described in more detail in the previous study (Yu et al., 2018). The field follows a winter wheat-summer maize rotation cropping system. Calcium superphosphate (150 kg P2O5 ha−1) and potassium sulphate (75 kg K2O ha−1) were applied before tillage. After each harvest, the total crop residue was directly shredded and incorporated into the soil through rotary tillage, to a depth of approximately 20 cm. The quantities of wheat (44.7% of total C, 0.76% total of N) and maize (38.6% of total C, 0.92% of total N) straw incorporated were approximately 3,000 kg ha−1 and 8,000 kg ha−1, respectively. This straw amendment practice began in 2010, with three replicate plots of 67 m2each.

Soil samples were collected in early November, 2016. In each replicate plot, five soil cores (3.0 cm in diameter) were randomly extracted from the plow layer and combined into a composite sample, resulting in a total of three soil samples. Each sample was sealed in a sterile plastic bag and transported to the laboratory on ice. After homogenization, the samples were sieved through a 2.0 mm mesh and stored at −80 °C until further analysis.

Total DNA was extracted from each soil sample using the PowerSoil DNA Isolation Kit (MoBio Laboratories, Carlsbad, CA, USA), following the manufacturer’s protocol with minor modifications (Yu et al., 2018). The extracted DNA was initially assessed using 1.0% agarose gel electrophoresis, and its purity was measured using a NanoDrop ND-2000 UV-VIS spectrophotometer (Thermo Fisher Scientific, Waltham, MA, USA). In addition, its concentration was determined using the Qubit dsDNA Assay Kit and a Qubit 2.0 Fluorometer (Thermo Fisher Scientific, Waltham, MA, USA). Finally, the DNAs extracted from the three soil samples were mixed at the same concentration in equal volumes for metagenomic sequencing.

Metagenomic sequencing and analysis

The pooled soil DNA was utilized for metagenomic library preparation and sequencing. Metagenomic libraries were constructed using the NEBNext Ultra DNA Library Prep Kit for Illumina (New England Biolabs, Ipswich, MA, USA). Briefly, the DNA was fragmented to a target size of 350 bp by sonication. The resulting fragments were end-repaired, ligated to full-length adaptors, and subsequently used to create paired-end sequencing libraries. High-throughput sequencing was performed on the Illumina HiSeq 2500 platform (Illumina, San Diego, CA, USA) at Novogene (Beijing, China), generating approximately 14.11 Gb of raw data. Low-quality reads containing more than 40 nucleotides with a quality score below 38, or more than 10 ambiguous bases (N), were filtered out. Additionally, adapter sequences were removed. After filtering, a total of 14.09 Gb of clean data was retained. These data were deposited in the NCBI Sequence Read Archive (SRA) under accession number PRJNA420606. The clean reads were assembled using MEGAHIT v1.1.1 with the “–presets meta-large” parameter, producing scaffolds (Li et al., 2015). These scaffolds were split at N-joins to generate scaftigs (continuous sequences without ambiguous bases). Only scaftigs ≥ 500 nt were retained for further open reading frame (ORF) prediction using MetaGeneMark v2.10 (Zhu, Lomsadze & Borodovsky, 2010). ORFs shorter than 100 bp were discarded, and the remaining sequences were clustered using CD-HIT v4.5.8 at a threshold of 95% identity and 90% coverage to construct a non-redundant gene catalog (Fu et al., 2012). The longest sequence within each cluster was selected as the representative for functional annotation. To assess gene abundance, the clean reads were realigned to the gene catalog using SoapAligner v2.21, with the parameters: -m 200, -x 400, identity ≥ 95% (Li et al., 2009). Genes with fewer than two mapped reads were excluded to minimize erroneous identifications. The relative abundance of each gene was calculated based on the number of reads mapped to the gene and the gene’s length.

For taxonomic assignment, predicted genes were aligned against the MicroNR dataset, which consists of microbial reference sequences from the NCBI non-redundant (NCBI-NR) database (Version: 2015-12-04), using DIAMOND v0.8.22 (BLASTp, e-value ≤ 1e−5) (Buchfink, Xie & Huson, 2015). Significant matches for each gene were defined as those with e-values ≤ 10 times the e-value of the top hit. Taxonomic classifications were determined using the lowest common ancestor algorithm implemented in MEGAN4 (Huson et al., 2011). The relative abundance of each taxonomic group was calculated by summing the relative abundances of the corresponding genes. For functional classification, DIAMOND v0.8.22 was used to align predicted genes with sequences in the Kyoto Encyclopedia of Genes and Genomes (KEGG) and Carbohydrate-Active Enzymes (CAZy) databases (BLASTp, e-value ≤ 1e−5). The best hits were selected for functional annotation.

Phylogenetic analysis of bacterial laccases

The deduced amino acid sequences from the predicted ORFs were screened to identify putative bacterial laccases using previously established profile hidden Markov models (pHMMs) (Ausec et al., 2011). Only sequences with an e-value below 1e−10 were retained. These sequences were manually validated against the NCBI Conserved Domain Database to confirm the presence of the characteristic cupredoxin domain (Dandare et al., 2019). The regions spanning between copper-binding regions I (cbr I) and II (cbr II) of the identified bacterial laccases were aligned with known sequences from the GenBank database using ClustalX2. A phylogenetic tree was constructed using the neighbor-joining method in MEGA v7.0, with bootstrap values calculated from 1,000 replications to assess the robustness of the tree topology (Kumar, Stecher & Tamura, 2016).

Cloning of bacterial laccase genes from soil DNA and sequence analysis

A subset of bacterial laccase genes was randomly selected for direct PCR amplification using soil DNA as the template. The amplified products were purified using the TIANgel Midi Purification Kit (Tiangen, Beijing, China) and subsequently cloned into pEASY-Blunt simple cloning vectors (TransGen, Beijing, China) for sequencing. The sequencing results were compared to the original sequences using BLAST for validation. Full-length cloning was performed on five selected laccase gene fragments. The 5′ and 3′ flanking regions were obtained using modified thermal asymmetric interlaced PCR TAIL-PCR (mTAIL-PCR), and these sequences were assembled with the known portions of the gene (Huang et al., 2010). The ORFs were identified using ORFfinder (https://www.ncbi.nlm.nih.gov/orffinder), and sequence identities were verified using BLAST. Signal peptides were predicted using SignalP 5.0 (http://www.cbs.dtu.dk/services/SignalP/), and multiple sequence alignments were generated using ClustalX2.

Heterologous expression and purification of lacS1 in Escherichia coli (E. coli)

The ORF of lacS1, excluding the signal peptide-coding region, was amplified from the soil metagenome using primers specifically designed for this study: ccatgg ATGCAACGCCGATTAGAAAGCTGTCG (italicized Nco I site) and gcggccgc CGCAACGCGCACGAGCGC (italicized Not I site). The PCR product was purified and cloned into a pEASY-Blunt simple cloning vector for sequencing. After verification, the lacS1 fragment was subcloned into the pET-28a(+) expression vector (Novagen; Merck KGaA, Darmstadt, Germany). The recombinant plasmid pET-lacS1 was then transformed into E. coli BL21 (DE3) competent cells.

Transformed E. coli BL21 (DE3) cells harboring pET28a-lacS1 were cultured at 37 °C in 1 L of Luria-Bertani (LB) medium containing 50 µg/mL kanamycin with agitation at 200 rpm. When the culture reached an OD600 of 0.5–0.6, enzyme expression was induced with 0.1 mM isopropyl-β-D-thiogalactoside and 0.25 mM CuSO4. To maximize the yield of soluble and active protein, agitation was stopped after 4 h of induction (28 °C, 150 rpm), and the cells were incubated under microaerobic conditions for an additional 20 h. The cells were harvested by centrifugation at 11,000 rpm (4 °C) for 15 min. The resulting cell pellets were resuspended in cold 20 mM Tris–HCl buffer (pH 7.6) containing 500 mM NaCl and 10% glycerol. Cell disruption was performed by sonication on ice using an ultrasonic cell disruptor (Scientz, Ningbo, China) with 3-s pulses and 5-s intervals. After centrifugation at 11,000× g for 30 min, the supernatant was subjected to affinity chromatography using a Ni2+-nitrilotriacetic acid (NTA) chelating column (Qiagen, Hilden, Germany). Elution was carried out using a stepwise gradient of imidazole (0, 20, 40, 60, 80, 100, 200, 300, and 500 mM) in 20 mM Tris–HCl, 500 mM NaCl, and 10% glycerol (pH 7.6). Fractions exhibiting enzymatic activity were collected, and the purity and molecular mass of the enzyme were assessed by sodium dodecyl sulfate-polyacrylamide gel electrophoresis (SDS-PAGE).

Characterization of lacS1

Laccase activity was measured using 2,2′-Azino-bis(3-ethylbenzothiazoline-6-sulfonic acid) (ABTS; Sigma-Aldrich, St. Louis, MO, USA) as the substrate. The reaction mixture contained 50 µL of appropriately diluted enzyme solution and 950 µL of 0.5 mM ABTS (ɛ420 = 36,000 M−1 cm−1) in 50 mM sodium citrate buffer (pH 5.0) supplemented with 100 µM CuSO4. The mixture was incubated at 50 °C for 5 min, followed by rapid cooling in an ice-water bath for 30 s to stop the reaction. Absorbance at 420 nm was measured spectrophotometrically. One unit (U) of laccase activity was defined as the amount of enzyme required to oxidize one µmol of ABTS per min. All assays were performed in triplicate.

The optimal pH for the purified recombinant lacS1 was determined by assaying laccase activity at 50 °C in 50 mM sodium citrate buffer (pH 3.0–6.0) and Na2HPO4-KH2PO4 buffer (pH 6.0–8.0). pH stability was assessed by measuring residual enzyme activity under standard conditions after incubation at various pH values (3.0–9.0) for 1 h at 37 °C. The buffers used for stability assays included 50 mM sodium citrate for pH 3.0–6.0, Na2HPO4-KH2PO4 for pH 6.0–8.0, and Tris–HCl for pH 8.0–9.0. Optimal temperature was determined by conducting the reaction at various temperatures (30–70 °C) in 50 mM sodium citrate buffer (pH 5.0). Thermal stability was evaluated by pre-incubating the enzyme at 40 °C and 50 °C for different durations, followed by activity measurements under standard conditions.

To assess the effect of CuSO4 on lacS1 activity, the purified enzyme was incubated with ABTS at the optimal temperature for 5 min in 50 mM sodium citrate buffer (pH 5.0) in the presence of 0–0.5 mM CuSO4. The effect of NaCl was similarly tested under standard conditions with varying concentrations.

Results

Overview of the metagenomic data analysis

To uncover bacterial laccases in the straw-amended soil, metagenomic sequencing was performed using the HiSeq 2500 platform. After quality control, approximately 99.86% of the reads were used for assembly, generating a total of 416,112 scaftigs. MetaGeneMark predicted 329,007 non-redundant ORFs from these sequences (Table S1).

Taxonomic analysis revealed that 308,011 (93.62%) of the predicted ORFs could be annotated using the microNR database. Bacteria were the most dominant group (88.11%), followed by Archaea (4.29%), Eukaryotes (0.09%), and Viruses (0.05%). At the phylum level, the most abundant bacterial groups (relative abundance ≥ 1%) were Proteobacteria, Actinobacteria, Acidobacteria, Gemmatimonadetes, Bacteroidetes, Thaumarchaeota, Nitrospirae, Chloroflexi, Cyanobacteria, Firmicutes, Planctomycetes, and Verrucomicrobia (Fig. 1).

Figure 1 Overview of taxonomic composition of the microbial community in the straw-amended soil.

Functional analysis of the microbial communities in the straw-amended soil was performed by comparing the predicted proteins from the metagenomic dataset to the KEGG database. The KEGG annotations indicated that carbohydrate metabolism was the most abundant functional category (Fig. S1). Additionally, the predicted genes were analyzed against the CAZy database to gain insights into the microbial degradation of plant polymers in the soil. Most of the identified CAZymes were categorized as glycoside hydrolases (GHs, 37.8%), followed by glycosyltransferases (GTs, 33.9%), carbohydrate-binding modules (CBMs, 23.9%), carbohydrate esterases (CEs, 6.8%), auxiliary activities (AAs, 4.2%), and polysaccharide lyases (PLs, 0.8%) (Table 1). The CAZyme repertoire consisted of 4,727 candidate GHs from 92 families, 4,221 GTs from 43 families, 2,980 CBMs from 38 families, 853 CEs from 14 families, 522 AAs from nine families, and 106 PLs from 12 families. These results suggest that the microbial consortia in the straw-amended soil possess substantial capabilities for plant biomass degradation.

Table 1 CAZymes classification of predicted ORFs from the straw-amended soil.

CAZymes classification	#ORF	%	
Glycoside hydrolases (GHs)	4,724	37.8%	
Glycosyltransferases (GTs)	4,221	33.8%	
Carbohydrate-binding modules (CBMs)	2,980	23.9%	
Carbohydrate esterases (CEs)	853	6.8%	
Auxiliary activities (AAs)	522	4.2%	
Polysaccharide lyases (PLs)	106	0.8%	
Total CAZymes*	12,489		
Notes.

* The total numbers of CAZymes is less than the sum (AAs+CBMs+CEs+GHs+GTs+PLs) due to the fact that some multimodular predicted proteins were detected.

Diversity and phylogenetic distribution of bacterial laccases in the straw-amended soil

To identify putative bacterial laccase-coding genes in the straw-amended soil metagenome, pHMMs were employed to search the deduced amino acid sequences of all predicted ORFs (Ausec et al., 2011). This analysis identified 322 putative bacterial laccases, including both two- and three-domain enzymes (Table 2). Notably, 85.4% of these putative laccases were retrieved using multiple models, likely due to the incomplete nature of many sequences.

Table 2 Number of putative bacterial laccases retrieved with pHMMs searching from the straw-amended soil metagenome.

Type	Model name	No. of genes found	
		Total	Unique	
Two-domain	typeB2D	185	5	
	typeC2D	142	11	
Three-domain	small3D	286	26	
	big3D	234	5	
	cot3D	108	0	
	more than one model	/	275	
		sum	322	
Notes.

/, not applicable; sum, number of genes retrieved in total.

To validate the metagenomic sequencing and assembly, 14 putative bacterial laccase genes were randomly selected for PCR amplification and sequencing. Among these, 12 PCR products were obtained with the expected sizes. Further sequencing showed that only one fragment had no significant identity with the corresponding predicted gene, while the remaining 11 fragments exhibited sequence identities above 96%, except for two sequences with 86% and 92% identity. These results suggest that most of the predicted laccase genes represent authentic laccase genes in the straw-amended soil microorganisms.

The amino acid sequences of the identified laccases were aligned with known sequences in the NCBI-NR, CAZy, NCBI environmental (NCBI-ENV), and Swiss-Prot databases to assess their similarity with known proteins (Fig. 2). BLASTp analysis against the NCBI-NR database showed that the sequence identities of the 322 putative bacterial laccases ranged from 41% to 97%, with an average identity of 70.22%. Approximately 24.84% of these sequences were most similar to proteins annotated as “hypothetical proteins” in NCBI-NR. None of the sequences exhibited high similarity (>95% sequence identity) to any entry in the CAZy database, suggesting that none of these sequences had been previously deposited in CAZy (Hess et al., 2011; Zhou et al., 2017). Additionally, 45 sequences with less than 30% identity were considered novel (Zhou et al., 2017; Pearson, 2013). Furthermore, only 4.66% of the putative laccases shared more than 75% identity with sequences in the NCBI-ENV database, indicating that these enzymes have not been widely identified in previous metagenomic studies (Hess et al., 2011; Zhou et al., 2017). Lastly, 48 sequences exhibited less than 30% identity to any known proteins in Swiss-Prot, further indicating that their activity has not been biochemically verified (Zhou et al., 2017). Collectively, these results highlight the novelty of the bacterial laccases identified in the straw-amended soil.

Figure 2 Similarity distribution of the putative bacterial laccsaes (n = 322).

Sequences were compared to the NCBI-NR (blue, 322 hits), CAZy (green, 194 hits), NCBI-ENV (red, 319 hits), and Swiss-Prot (purple, 289 hits) databases (best BLAST hit, E value ≤1e−5).

The phylogenetic origins of the identified laccase genes were further explored through taxonomic classification (Fig. 3). These genes were found to be unevenly distributed across various phyla, with the majority originating from Proteobacteria (37.21%), Actinobacteria (16.32%), and Gemmatimonadetes (13.50%). Additional phyla contributing to laccase production included Thaumarchaeota (7.47%), Bacteroidetes (3.43%), Firmicutes (2.81%), Chloroflexi (1.53%), Crenarchaeota (1.08%), and Armatimonadetes (1.05%). Other phyla such as Acidobacteria, Cyanobacteria, Planctomycetes, Acetothermia, and Nitrospirae were also detected, albeit in lower relative abundances. At the order level, the genes were distributed across 42 different orders, with 17 orders representing over 1% of the total abundance (Fig. 3). The highest number of laccase-coding genes was affiliated with Gemmatimonadales (13.50%), followed by Geodermatophilales (6.86%), Nitrososphaerales (4.97%), and Burkholderiales (4.37%) (Fig. 3).

Figure 3 Hierarchical classification and distribution of the putative bacterial laccases in the straw-amended soil using KRONA extension.

To further explore the diversity and relationship of bacterial laccases in the straw-amended soil, amino acid sequences spanning between cbr I and cbr II of the putative laccases were used to construct a phylogenetic tree. Therefore, 74 putative laccases were selected, and 29 representative sequences from 20 genera and environmental samples were used as references. The bacterial laccase fragments were separated into seven distinct clusters (A–G) (Fig. 4). The laccase fragments from the straw-amended soil were found in all seven clusters, underscoring their substantial diversity. Cluster A and Cluster E were the most populated, containing 23 and over one-third of the bacterial laccase fragments from the soil, respectively. Cluster B contained three laccases with no known references, suggesting high novelty. Clusters C and F had two bacterial laccase fragments from the soil and 16 reference sequences, while seven bacterial laccase fragments from the straw-amended soil and a reference sequence from giant panda feces were grouped into cluster D. Cluster G contained 12 bacterial laccase fragments associated with various genera of Proteobacteria, such as Pantoea, Klebsiella and so on, indicating high diversity of laccase-producing Proteobacteria in the straw-amended soil.

Figure 4 Phylogenetic analysis of bacterial laccases in the straw-amended soil.

The lengths of the branches indicate the relative divergence among the amino acid sequences. Scale bar represents 0.10 amino acid substitution per position. The numbers at the nodes indicate bootstrap values based on 1,000 bootstrap replications and bootstrap values (>50) are displayed. The reference sequences are marked with a closed circle (•) with GenBank accession numbers in parentheses.

Cloning and expression of bacterial laccase genes from the straw-amended soil

The complete ORFs of five representative bacterial laccase genes were obtained using mTAIL-PCR (Table S2). Together with two full-length genes directly cloned from the soil metagenome, these seven bacterial laccases were translated into amino acid sequences. These seven bacterial laccases consisted of 445–629 amino acids, with molecular weights ranging from 49 kDa to 70 kDa and theoretical isoelectric points between 5.5 and 9.3. Among them, S_14142, S_63994, S_326900, and S_507105 contained TAT signal peptides, indicating their potential secretion via the twin-arginine translocation system (Table 3 and Fig. 5). Sequence alignments revealed low sequence identity among the seven laccases, with identities ranging from 23% to 52%, except for S_230270 and S_394411, which shared 70% identity (Table S3). Despite the differences at the whole protein level, the copper-binding sites were highly conserved (Fig. 5). None of the laccases exhibited more than 80% identity to previously reported sequences, confirming their novelty (Table S2).

Table 3 Sequence characteristics of the bacterial laccases.

Laccase	Protein size (amino acids)	Molecular weight	Theoretical pI	Signal peptide	
S_14142	503	55.9 kDa	8.62	Tat	
S_23069	461	50.9 kDa	6.18	No	
S_63994	445	49.1 kDa	7.97	Tat	
S_230270	629	69.5 kDa	5.51	No	
S_326900	542	59.1 kDa	9.29	Tat	
S_394411	553	61.7 kDa	5.72	No	
S_507105	575	63.8 kDa	6.20	Tat	

Figure 5 Multiple amino acid sequence alignment of different bacterial laccases.

Highly and moderately conserved amino acids are highlighted by black and grey boxes, respectively; TAT signal sequences are underlined; copper binding sites are marked by red frames. The protein sequences used for alignments were Lac15 from marine metagenome (ADM87301), X. arboricola laccase (AAA72013), T. terrenum laccase (WP_012874485) and Thioalkalivibrio ThioLacc (CCV01628).

For functional validation, S_63994, the most abundant gene, was designated lacS1 and expressed in E. coli without the signal peptide-coding sequence. Following microaerobic induction, recombinant lacS1 was purified using Ni2+-NTA metal-affinity chromatography, yielding a single protein band with an apparent molecular mass of approximately 48 kDa on 10% SDS-PAGE (Fig. S2). To verify the identity of the purified enzyme, the target band was excised and analyzed by LC-ESI-MS/MS, revealing eight peptides matching the recombinant lacS1 sequence and confirming its identity (Fig. S3).

For functional characterization of the recombinant lacS1, ABTS was used as the substrate. The enzyme was active across a broad pH range, spanning from 4.0 to 8.0, with optimal activity observed at pH 5.0 (Fig. 6A). It retained over 60% of its maximum activity after incubation at 37 °C for 1 h across pH values ranging from 5.0 to 9.0, demonstrating significant stability within this range (Fig. 6B). The optimal temperature for lacS1 activity was found to be 50 °C, and the enzyme exhibited thermostability at 40 °C (Figs. 6C–6D). However, lacS1 showed a gradual loss of activity when incubated at 50 °C for prolonged periods (Fig. 6D). The addition of copper ions to the reaction mixture had no significant effect on the enzyme’s activity (Fig. 7A; p < 0.05 by Tukey). In contrast, increasing concentrations of NaCl gradually reduced lacS1 activity, with the enzyme retaining over 42% activity at 250 mM NaCl. However, at concentrations exceeding 500 mM NaCl, the enzyme rapidly lost activity (Fig. 7B).

Figure 6 Characterization of recombinant lacS1.

(A) Effect of pH on activity of recombinant lacS1. (B) pH stability of recombinant lacS1. (C) Effect of temperature on activity of recombinant lacS1. (D) Thermostability of recombinant lacS1.

Figure 7 Effect of CuSO4 (A) and NaCl (B) on activity of recombinant lacS1.

The different letters on bars indicate the means differ significantly according to one-way ANOVA followed by Tukey’s multiple range test (P < 0.05).

Discussion

In this study, we utilized metagenomic sequencing to explore bacterial laccases in the straw-amended soil. Metagenomics is a powerful approach for discovering novel biocatalysts that are often inaccessible through traditional pure culture methods. By constructing metagenomic libraries and employing sequence- or activity-based screening, numerous bacterial laccases with promising industrial applications have been identified in previous studies (Ye et al., 2010; Ausec et al., 2017; Fang et al., 2011; Fang et al., 2012; Yang et al., 2018). However, metagenomic library construction and screening can be labor-intensive and typically yield limited information (Robinson, Piel & Sunagawa, 2021). In contrast, high-throughput sequencing without the need for cloning allows for a more comprehensive analysis of microbial community structure and function, enabling a more thorough investigation of bacterial laccases in environmental samples, such as straw-amended soil.

We first analyzed the microbial communities in the straw-amended soil. The results indicated that the dominant bacterial phyla were consistent with those identified in our previous study using 16S rRNA gene amplicon sequencing (Yu et al., 2018). The only notable differences were the detection of Thaumarchaeota and Cyanobacteria as dominant phyla in this study. These discrepancies may arise from the limitations of 16S rRNA gene amplicon sequencing, which is influenced by the variability of 16S rRNA gene sequences and copy numbers in bacterial genomes (Větrovský & Baldrian, 2013). Additionally, the primer pair used in the previous study (original 515F and 806R) has been shown to exhibit a bias against Thaumarchaeota (Hugerth et al., 2014).

Based on the metagenomic sequencing data, bacterial laccases were identified using laccase-specific pHMMs, rather than simple BLAST searches, to improve accuracy (Ausec et al., 2011). Laccases belong to the MCO superfamily and are difficult to discriminate from other MCOs. Furthermore, many of the ORFs in our dataset were incomplete, necessitating the use of pHMMs for a more accurate identification. Although metagenomic sequencing and assembly have inherent error rates, PCR amplification and Sanger sequencing validated the authenticity of most of the bacterial laccase genes identified in our dataset.

The phylogenetic distribution of laccase-coding genes at the phylum level closely mirrored the overall microbial community structure in the straw-amended soil. This supports the idea that laccase gene composition may be linked to factors that structure microbial communities or that there is a correlation between the functional traits of microbial consortia and their taxonomic profiles (Zhou et al., 2017; Lauber, Sinsabaugh & Zak, 2009; Wang et al., 2016). At the order level, Gemmatimonadales, which contained the largest number of laccase-coding genes, is part of the Gemmatimonadetes phylum and is believed to play a role in phosphorus removal (Wang et al., 2009). Although Gemmatimonadales has few cultivable representatives, studies suggest that it possesses high sulfur and nitrogen metabolic diversity (Rasigraf et al., 2020). Furthermore, it has been found to be more abundant in soils polluted with aromatic compounds, indicating its potential role in aromatic compound degradation (Cecotti et al., 2018; Thelusmond et al., 2018). Therefore, Gemmatimonadales may play a central role in SOM and nutrient cycling, contributing to improved soil fertility and plant growth. In addition to Gemmatimonadales, orders such as Geodermatophilales, Micrococcales, Jiangellales and Rubrobacterales of the Actinobacteria phylum, and Nitrososphaerales of the Thaumarchaeota phylum were also represented. Actinobacteria are usually found as a major component of bacterial consortia with high lignolytic activity (Wang et al., 2013; Moraes et al., 2018). Thaumarchaeota include all known ammonia-oxidizing archaea and multiple laccases have been identified in Nitrososphaerales genomes (Kerou et al., 2016). Moreover, several orders within the Proteobacteria phylum, including Burkholderiales, Rhodospirillales, Xanthomonadales, Rhizobiales, Myxococcales, Hydrogenophilales, Pseudomonadales, and Desulfuromonadales, were also observed as major laccase-producers in the straw-amended soil. This is in line with a recent study conducted in coniferous forest soils from across North America, which identified many of these taxa as among the top 10 richest laccase-coding gene-harboring bacteria (Wilhelm et al., 2019). The laccase profile reveals that the predominant Proteobacteria cooperate with Actinobacteria and Gemmatimonadales to synergistically act on the decomposition of lignin.

A phylogenetic tree was constructed using the bacterial laccase fragments containing the conserved regions between cbr I and cbr II, along with reference sequences. This analysis further demonstrated the significant diversity of bacterial laccases in the straw-amended soil, many of which may exhibit valuable enzymatic properties. For example, reference sequences in cluster A were derived from three marine bacterial laccases. These marine laccases exhibit strong chloride tolerance, with Lac21 maintaining its original activity in the presence of 250 mM NaCl, and Lac15 and Lac1326 maintaining their original activity even at 1 M NaCl (Fang et al., 2012; Fang et al., 2011; Yang et al., 2018). The close phylogenetic relationship between the bacterial laccase fragments from the straw-amended soil and these reference sequences suggests the presence of Cl− resistant bacterial laccases. The bacterial laccase fragment in cluster C was closely related to laccases from Thermus thermophilus, Meiothermus ruber, and Sinorhizobium meliloti, which have been demonstrated to possess high thermostability, suggesting similar properties for this enzyme (Miyazaki, 2005; Kalyani et al., 2016; Pawlik et al., 2016).

Furthermore, seven bacterial laccase genes with full-length ORFs were translated into amino acid sequences, and four of these contained TAT signal peptides, indicating bacterial laccases could be involved in extracellular lignin degradation through the twin-arginine translocation system, in addition to various intracellular developmental processes. Characterization of lacS1 revealed it was most active at slightly acidic pH when ABTS was used as the substrate. This behavior is similar to the laccase from Geobacillus yumthangensis; however, lacS1 demonstrated greater stability under basic conditions, akin to CotA laccases from Bacillus species (Sharma & Leung, 2021; Liu et al., 2023). Additionally, lacS1 showed optimal activity at 50 °C but remained stable only up to 40 °C, suggesting it is a mesophilic enzyme. Notably, the addition of copper ions did not affect the enzyme’s activity, likely due to the incorporation of Cu2+ into the active sites during laccase synthesis under microaerobic fermentation conditions (Sun et al., 2024). Moreover, lacS1 displayed halotolerance, similar to LacM, though weaker than the marine laccases Lac15, Lac21, and Lac1326 (Fang et al., 2012; Ausec et al., 2017; Fang et al., 2011; Yang et al., 2018). These differences in halotolerance may be attributed to the distinct habitats of the laccase-producing bacteria. Overall, these properties suggest that lacS1 is well-suited for applications in wastewater treatment.

It is important to highlight that the findings regarding bacterial laccase diversity in this study are limited to the specific sampling location. This is due to the distinct responses of soil microbial communities to straw incorporation, which vary based on factors such as soil taxonomy, soil properties, straw materials, and fertilization practices. In this study, we focused specifically on mining bacterial laccases using the straw-amended soil as material. Moving forward, other cloned bacterial laccase genes will be heterologously expressed to determine their enzymatic properties. Through a combination of mutagenesis experiments and functional studies, their catalytic mechanisms will be elucidated. This will pave the way for the development of novel bacterial laccases with enhanced performance and significant potential for diverse applications.

Conclusions

In this study, we investigated bacterial laccases in a straw-amended soil using metagenomic sequencing and a pHMM-based search. The identified bacterial laccases were predominantly distributed across the Proteobacteria, Actinobacteria, and Gemmatimonadetes phyla, indicating their capacity for laccase production to synergistically act on the decomposition of lignin. These laccases exhibited substantial novelty and diversity, with many likely possessing promising enzymatic properties. The characterization of recombinant lacS1 revealed good pH stability and chloride tolerance, indicating its potential for application in wastewater treatment. This study provides valuable materials for the development and exploitation of bacterial laccase resources.

Supplemental Information

Supplemental Information 1 Supplemental Tables

Supplemental Information 2 Sequence data

Supplemental Information 3 Raw data: characterization of lacS1

We extend our sincere thanks to Xiaojun Song, Qimeng Wang, and Fangfang Pan for their technical support. We are also grateful to Haijie Yan, Zhiguo Wen, and Xiumei Li for their insightful discussions and contributions to this research.

Additional Information and Declarations

Competing Interests

Author Contributions

DNA Deposition

Data Availability

The authors declare there are no competing interests.

Dali Yu conceived and designed the experiments, performed the experiments, analyzed the data, prepared figures and/or tables, authored or reviewed drafts of the article, and approved the final draft.

Ying Liu analyzed the data, prepared figures and/or tables, and approved the final draft.

Hongying Cai performed the experiments, prepared figures and/or tables, and approved the final draft.

Wanqiu Huang performed the experiments, prepared figures and/or tables, and approved the final draft.

Huijun Wu analyzed the data, authored or reviewed drafts of the article, and approved the final draft.

Peilong Yang conceived and designed the experiments, authored or reviewed drafts of the article, and approved the final draft.

The following information was supplied regarding the deposition of DNA sequences:

The nucleotide sequences of the seven full-length bacterial laccase genes are available at Genbank: PQ350336–PQ350342.

The following information was supplied regarding data availability:

The sequences are available at NCBI SRA: PRJNA420606.

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
