# Peer review of "Metagenomic investigation of bacterial laccases in a straw-amended soil"

_PeerJ, doi:10.7717/peerj.19327_

## Round 0.1 · original submission · Major Revisions

Dear authors,

I have gone through your manuscript. It is interesting, but there are several flaws throughout the manuscript which were recognized by the reviewers. Please carefully revise your manuscript taking all issues raised by the reviewers into account.

Kind regards,
Elisabeth Grohmann

·

Basic reporting

I have carefully reviewed the manuscript entitled “Metagenomic investigation of bacterial laccases in a straw-amended soil” by Yu et al. submitted to the journal PeerJ. Laccases hold significant promise as environmentally friendly biocatalysts for various industrial applications. Most identified and characterized laccases originate from fungal sources, and fungal laccases are extensively used in industrial processes due to their relatively high activity. However, continuously increasing evidence has shown that laccases are also prevalent in bacteria and meanwhile, bacterial laccases exhibit several advantageous properties, such as high thermal stability, chloride tolerance, and broad pH stability, making them more suitable for use in harsh industrial environments, by comparison with their fungal counterparts. Despite their potential, only a limited number of bacterial laccases have been thoroughly characterized, underscoring the need for further investigation into their properties for novel biotechnological applications. The straw-amended soil serves as a unique habitat for lignocellulosic-degrading microbial consortia, it is likely to be rich in bacterial laccase resources. Therefore, this study comprehensively investigated the straw-amended soil through metagenomic sequencing to uncover novel laccases with potential for biotechnological applications. In my opinions, the research purpose is clear.

Experimental design

The experimental design is correct and testing methods used here are also reasonable. Besides, the manuscript is well-structured and well-written.

Validity of the findings

Overall, this study provided some of useful data and these data could advance the understanding of laccases-producing soil microbial ecology and offer the direction towards the excavation of soil bacterial laccases.

Additional comments

However, there are some small problems in the manuscript and these problems must be addressed before publication. The problems are listed as follows:
1. The soil background information should be supplemented, including soil type, texture and physicochemical properties. The information involved in straw material should be added, particularly nutrient (N, P, K) concentration. In addition, the information associated with fertilization practices should also be added. There exist distinct responses of soil microbial community to straw incorporation under conditions of various soil types, straw materials and fertilization patterns. This study only takes the soil samples at a single site and thus is necessary to tell readers the information involved.
2. In discussion part, authors should clearly state the shortcomings of this study, namely, all results observed here are from a given research site (a single soil). Meanwhile, authors should clearly state their next works.
3. Commonly, the control treatment without straw amendment is included in field experiment. Why wasn't the soil without straw amendment collected for bacterial laccase investigation? Maybe, the difference in bacterial laccase between the control and straw incorporation are more interesting for readers and can more powerfully guide the excavation of soil bacterial laccases.

·

Basic reporting

The submitted manuscript entitled “Metagenomic investigation of bacterial laccases in a straw-amended soil” studied the effects of long-term wheat and maize straw as soil amendment on soil and soil bacterial communities and genes related to bacterial laccases. The topic in this study is suitable for the aim of the journal of PeerJ and important subject for understanding the source on isolating bacterial laccases in the future for industrial used. This paper is well presented and discussed. However, there are several issues to understand the valuable data and discussion in this study. I recommend authors revise it after considering the following.

The introduction is well-written, presenting clear and coherent ideas. However, it lacks information on whether most of the bacterial laccase are cultured or uncultured bacteria. To strengthen this section, the authors should include relevant literature in the final paragraph (Lines 109–119) to clarify this distinction. Highlighting this distinction will better support the choice of the metagenomic approach and reinforce the study's objectives.

Experimental design

1. The study does not clearly specify whether the straw used was fresh or composted in Line 126-129. Additionally, chemical properties of the straw, particularly the C-organic content, are not provided. To enhance clarity and reproducibility, the authors should specify the type of straw used and include its relevant chemical properties.

2. The authors clearly state that the soil sample was compositely collected from five cores in Line 131-133. However, it is unclear whether these cores were taken from a single plot or multiple plots. Additionally, the number of replicates used for the metagenomic analysis is not specified. To improve clarity, the authors should clarify and provide details the total number of samples analyzed.

3. Is the DNA concentration mentioned in Lines 140–142 referring to the extracted DNA? Please clarify this point for better understanding.

4. Regarding the metagenomic analysis mentioned in Lines 146–150, did the authors amplify the DNA prior to HiSeq sequencing? If so, please clarify this and provide details on the primer set used, as well as the targeted 16S rRNA regions in this study.

5. Please provide the reference(s) for the primer set used to amplify the ORF of lacS1 (Lines 205–206).

Validity of the findings

Data and Results:
1. The metagenomic data analysis Lines 262-265 showed that the classification values decrease from the order to species level. Could you clarify why this occurs and provide more context?

2. The authors mention that 14 putative bacterial laccase genes were randomly selected to validate the metagenomic sequencing in Lines 292-293. Is this sample size sufficient to represent the entire ecosystem? Please elaborate.

3. The results of described of authentic laccase genes in Lines 294-298 are not linked to any table or figure. Please indicate the corresponding table or figure to help readers locate the data.

4. When mentioning 17 orders of laccase genes in Line 322, please specify the relevant table or figure in parentheses for easier reference.

5. The values for laccase-coding genes in the text (Lines 322-325) differ from those in Figure 3. Please ensure consistency between the text and the figure.

6. The phyla-genera text in Figure 3 is difficult to read. Consider increasing the font size and providing a higher-resolution version.

7. The paragraph in Lines 314-325 lacks a clear statement highlighting the significance or novelty of the findings. Please add a sentence to emphasize this in the concluding part.

8. Remove "et al." in Line 325 and specify the table or figure containing the data.
9. The authors mention using 28 representative sequences in Lines 328-329, but Figure 4 shows 29. Please double-check and revise accordingly.

10. The references used for phylogenetic tree analysis are stated to belong to 20 genera in Lines 328-329. Have these genera been previously reported to produce bacterial laccases? Please clarify.

11. The authors separated bacterial laccase fragments into seven distinct clusters in Line 330. Could you explain the criteria or method used to distinguish these clusters?

12. Five representative bacterial laccase genes were identified using mTAIL-PCR. Please indicate the table or figure showing this result.

13. The text states a range of 450–630 amino acids in Lines 344-346, but Table 3 shows 445–629. Please verify the correct range and reference the table in the manuscript.

14. Specify the table or figure containing the relevant data in Lines 349-350 and 352.

15. After mentioning “across a broad pH range,” in Line 361, please include the specific pH range (pH 5–8) for clarity.

16. The authors mention that copper ions had no significant effect on enzyme activity in Line 366-367. Was statistical analysis conducted to confirm this? Please clarify.

17. Revise the Figure 7A and 7B to include error bars and indicate the results of the statistical analysis.

18. What do the “/” and “sum” in Table 2 represent? Please explain or revise for clarity.

Discussion and Conclusion:
1. Please provide the appropriate reference(s) to support the statements made in Lines 378-379.

2. The authors mention the role of the Gemmatimonadetes phylum in phosphorus removal in Lines 402-404. How does this relate to laccase bacterial activity? Please clarify this connection.

3. Replace “So” with “Therefore” for a more formal tone in Line 407.

4. Ensure there is a space between “laccases” and “have” in Line 413.

5. The authors discuss the predominant phyla and orders related to laccase genes in Lines 398–421 but do not emphasize which laccase gene is most abundant in the straw-amended soil. Concluding this paragraph with a clear statement highlighting the most abundant laccase gene would provide better clarity and strengthen the discussion.

6. The authors have not yet discussed the novel laccase genes identified in this study. Adding a dedicated paragraph to emphasize these novel findings would strengthen the manuscript’s originality and significance.

Reviewer 3 ·

Basic reporting

Clear, unambiguous, professional English language used throughout.
The manuscript would benefit from a thorough revision by a native English speaker to enhance clarity and flow. For example:
• The abstract contains essential information but would be more effective if reorganized into logical sections: background/context, methods, overall results, specific findings on the recombinant enzyme (rlacS1), and implications/conclusions. Currently, the description of rlacS1 is interspersed with general information about bacterial laccases, which might confuse readers.
• Repetition of details about rlacS1 (e.g., “optimum temperature of the recombinant enzyme was 50°C” and “rlacS1 exhibited high Cl⁻ tolerance”) detracts from readability. Consolidating these details into a single cohesive description would reduce redundancy and improve the narrative.
Grammar, Precision, and Conciseness
• Grammar:
o Original: "Bacterial laccases was comprehensively investigated..."
 Correction: Replace was with were to ensure subject-verb agreement.
o Repeated use of “In addition” in consecutive sentences (e.g., “In addition, the optimum temperature...”) disrupts flow. Consider rephrasing for variety and smoother transitions.
• Precision and Conciseness:
o Original: “These findings provide valuable insights into the discovery of bacterial laccase resources and offer potential materials for their modification and industrial utilization.”
o Revised: “These findings highlight novel bacterial laccase resources with potential for industrial applications and enzyme engineering.”
Novelty and Impact
The statement that the laccases exhibit "significant novelty and diversity" could be strengthened with specific examples. Emphasize the industrial relevance of these findings, such as unique properties of rlacS1 or other laccases identified. Highlighting their potential for bioremediation or industrial applications would enhance the impact.
* * *
2. INTRODUCTION AND BACKGROUND
The introduction is comprehensive and well-structured, providing relevant context. However, revisions could enhance clarity and precision. For example:
• Explicitly define how the study fills a knowledge gap.
• Ensure smooth transitions between the discussion of laccase diversity, their enzymatic properties, and their industrial/bioremediation potential.
* * *
3. LITERATURE WELL-REFERENCED & RELEVANT
The references are relevant but outdated, with the most recent cited works from 2020. Significant advancements in this field have been made over the past four years. Including more recent publications would improve the paper’s credibility and alignment with current research trends.
* * *
4. STRUCTURE
The manuscript adheres to PeerJ standards and discipline norms. However, further clarity could be achieved through revisions as outlined above.
* * *
5. FIGURES
The figures are relevant and well-labeled, but Figure 3 requires improvement in resolution and quality for better readability.
* * *
6. RAW DATA
The raw data has been supplied and appears to meet PeerJ’s policy requirements.

Experimental design

Original Research
The study fits within the scope of the journal.
Research Question and Methods
The research question is well-defined, addressing a relevant gap. However, the methods require more detail to allow replication. Specific suggestions include:
• Expanding the characterization of rlacS1 to strengthen its relevance for bioremediation. For instance, testing the enzyme’s ability to degrade dyes or xenobiotic compounds would enhance its significance.
• Including statistical analyses to substantiate claims and ensure robust results.

Validity of the findings

The study employs metagenomic sequencing and a pHMM-based search to analyze laccases in straw-amended soil. The identified bacterial laccases predominantly belong to Proteobacteria, Actinobacteria, and Gemmatimonadetes phyla. The novelty and diversity of these laccases are highlighted, but additional experiments could strengthen the findings.
Suggestions:
• Conduct experiments to explore rlacS1’s potential for bioremediation, such as its capacity to degrade dyes or other xenobiotic compounds.
• Compare the findings more comprehensively with recent literature to underscore the novelty of the results.
Conclusions
The conclusions align with the research question but lack robustness. They should be refined to emphasize the potential of rlacS1 for wastewater treatment, supported by additional experiments or analysis.

Additional comments

no comment

---

## Round 0.2 · Minor Revisions

Your manuscript has highly improved by the revision. However, there are still some issues raised by Reviewer 2 which should be fixed. Please revise these minor issues carefully.

·

Basic reporting

This revised manuscript is well written, and the authors have effectively addressed all reviewer comments. I believe the article is close to acceptance, pending minor revisions in certain aspects of the experimental design, results, and discussion. Additionally, please carefully proofread the manuscript, as I still found some typographical errors.

Experimental design

The authors have clarified that they used their original primer sets to amplify the lacS1 ORF. It would be better if they explicitly include this information in the manuscript.

Validity of the findings

The authors have clarified the statistical analysis used in their study. I recommend explicitly stating the statistical method and significance level in the relevant figures or tables where significance results are discussed in the manuscript.

For example, in Lines 366–367 (revised manuscript):

"The addition of copper ions to the reaction mixture had no significant effect on the enzyme’s activity (Figure 7A; p > 0.05 by Tukey)."

Reviewer 3 ·

Basic reporting

no comment

Experimental design

no comment

Validity of the findings

no comment

Additional comments

From my perspective, all issues have been addressed. The manuscript is now well-written and up to date, with nothing to add in the revision. I recommend its approval.

---

## Round 0.3 · accepted · Accept

You have done a good job carefully revising the manuscript. Congratulations for the acceptance of your manuscript!